# Chronic Lymphocytic Leukemia: Investigation of Survival and Prognostic Factors with Drug-Related Remission

**DOI:** 10.3390/diagnostics15060728

**Published:** 2025-03-14

**Authors:** Gökhan Pektaş, Ercan Gönül, Şeyma Öncü, Merve Becit Kızılkaya, Gökhan Sadi, Mehmet Bilgehan Pektaş

**Affiliations:** 1Division of Hematology, Faculty of Medicine, Muğla Sıtkı Koçman University, 48000 Muğla, Türkiye; gokhanpektas@gmail.com (G.P.); ercan.gonul93@hotmail.com (E.G.); 2Department of Medical Pharmacology, Faculty of Medicine, Afyonkarahisar Health Sciences University, 03200 Afyonkarahisar, Türkiye; seyma.oncu@afsu.edu.tr; 3Department of Toxicology, Faculty of Pharmacy, Afyonkarahisar Health Sciences University, 03200 Afyonkarahisar, Türkiye; merve.becit@afsu.edu.tr; 4Department of Biology, K.O. Science Faculty, Karamanoglu Mehmetbey University, 70100 Karaman, Türkiye; sadi.gokhan@gmail.com

**Keywords:** drug-related remissions, chronic lymphocytic leukemia, Binet, Rai, 17p deletion

## Abstract

**Background/Objectives:** Understanding the pathogenesis of chronic lymphocytic leukemia (CLL) has led to the development of new prognostic and diagnostic tools, and efforts are underway to extend survival with new prognostic markers and treatment agents. This study aims to evaluate the factors affecting the prognosis and survival of patients with CLL via a retrospective study. **Methods:** Accordingly, the demographic features of, clinical and laboratory findings for, and hematological parameters and treatment responses of 178 CLL patients who were followed between 1 January 2015 and 31 December 2024 were analyzed before and after treatment protocols were carried out. **Results:** During the follow-up period, 40.8% of the patients received medical therapy, with 42.5% achieving complete remission, 49.3% experiencing partial remission, and 8.2% demonstrating no response to the treatments. The results demonstrated that an advanced Binet stage, the presence of splenomegaly, a positive direct Coombs test, the presence of a 17p deletion, thrombocytopenia, and elevated creatinine, leukocyte, and lymphocyte counts were associated with increased mortality. Elevated Binet and Rai stages, the existence of 17p deletion, and reduced hemoglobin levels were identified as statistically significant factors. **Conclusions:** Given the unfavorable prognosis of CLL patients exhibiting a positive direct Coombs test and compromised renal function, further investigations are required to validate the necessity of more rigorous monitoring and, possibly, early intervention. These findings underscore the importance of identifying high-risk factors in CLL to optimize patient management and improve long-term outcomes.

## 1. Introduction

Chronic lymphocytic leukemia (CLL) is a clonal neoplasia defined by the presence of blastic cells consisting of CD5+ mature B lymphocytes in the peripheral circulation. It is more common in older people, and the average age of diagnosis is 70, with males being twice as likely as females to have CLL [1]. The 65-to-74 age group exhibits the most cases [2], and 95% of people who are diagnosed with CLL also have at least one other health problem [3]. It is the most common type of leukemia in adults, with a prevalence varying by race and geographic region [4]. In the United States, CLL accounts for 1% of all new cancer cases, with a reported five-year survival rate of 88% [3], and it is estimated to cause 4410 deaths annually. There is a hereditary genetic predisposition to CLL, with family members having a 6- to 9-fold-higher risk [5]. Additionally, exposure to pesticides and herbicides has been associated with the development of this disease [6,7]. Chronic lymphocytic leukemia is diagnosed by the presence of mature B lymphocytes in the peripheral blood and characterized by B cell markers and the CD5 antigen, with a count exceeding 5000/mm^3^ and fewer than 55% prolymphocytes [8]. The disease generally originates from specific genomic alterations that disrupt the regulation of proliferation and apoptosis in clonal B cells, which co-express surface membrane immunoglobulin (smIg) along with low levels of Ig light chain, CD79b, CD20, and CD23 [9]. The prognosis of CLL varies significantly, likely due to a combination of environmental factors and genetic and epigenetic changes. One key determinant of prognosis is the mutation status of the immunoglobulin heavy-chain variable region (IgHV) gene, which is also associated with significantly prolonged survival [10]. Hypogammaglobulinemia, trisomy 12, del(13q), del(11q), del(17p), and TP53 mutations have been shown to be associated with disease progression and survival [11]. Furthermore, direct Coombs positivity is a risk factor that cannot be captured by classical staging systems, especially for early-stage patients, while 17p deletion is a strong prognostic marker in all stages and crucial in defining the “high molecular risk” subgroup, especially for early- and intermediate-stage patients. Additionally, numerous potential biomarkers have emerged in recent years. Poor prognostic indicators include age, male gender, a short lymphocyte doubling time, CD38 positivity (>30%), genetic deletions, high ZAP70 expression, elevated serum lactate dehydrogenase (LDH) levels, diffuse bone marrow involvement, and a short response time to treatment [12]. The clinical course of CLL is highly heterogeneous; while one-third of patients require treatment upon diagnosis, another third may live without ever needing it, and the remaining third will require treatment at some point during their lifetime. As a result, prognostic staging is crucial for determining which patients will need early intervention and which treatment strategies will be most beneficial [13]. The modified Rai and Binet systems are widely used, cost-effective, and internationally accepted staging methods for evaluating, diagnosing, and predicting the prognoses of CLL patients in both routine practice and clinical settings [11]. The Rai system is based on the levels of lymphocytosis, lymphadenopathy, splenomegaly, anemia, and thrombocytopenia and comprises four stages [14,15]. Similarly, the Binet staging system (A, B, and C) considers the presence of lymph nodes, organomegaly, anemia, and thrombocytopenia [15,16]. While these systems have served as the foundation for other prognostic models, they have been found to fall short in identifying aggressive disease at early stages [17]. This limitation is attributed to advancements in medical techniques, which have revealed a broader range of biological and molecular factors influencing prognosis. These factors include IgHV gene mutations, cytogenetic abnormalities, aberrant protein expression, and the characteristics of the host immune environment, all of which impact prognosis regardless of the stage of disease. Currently, more than 80% of CLL patients are diagnosed at an early stage with a low tumor burden according to the Rai and Binet systems. However, these criteria remain insufficient for accurately predicting disease progression [18]. Additional prognostic information is necessary to account for the heterogeneity of the CLL population and distinguish patients with favorable outcomes from those with poor prognoses.

The aim of this study is to evaluate the demographic, laboratory, and clinical characteristics of patients diagnosed with CLL who were treated at tertiary health institutions between 2015 and 2024 and assess how these factors influenced prognosis. Additionally, this study examines the impact of various drugs on patient outcomes. By analyzing these aspects, we seek to enhance our understanding of CLL progression and treatment efficacy, ultimately contributing to improved patient management and personalized therapeutic strategies.

## 2. Materials and Methods

### 2.1. Study Design and Sample Information

A descriptive, cross-sectional, retrospective study was conducted to assess data on patients diagnosed with CLL at Muğla Sıtkı Koçman University Hospital, Division of Hematology, from 1 January 2015 to 31 December 2024, encompassing all inpatients treated within this timeframe, without selecting a specific sample.

### 2.2. Data Collection and Study Variables

Demographic characteristics of the patients, including gender and age, as well as the presence of malignancy, splenomegaly, and lymph node involvement, were recorded. Other data collected included the duration of hospitalization, prognosis (mortality), and laboratory findings (hemogram, IGG, sedimentation rate, AST, ALT, LDH, creatinine, uric acid, albumin, total protein, and total and direct bilirubin). Additionally, the direct Coombs test and genetic analyses (Trisomy 12, 17p deletion, and 13q deletion) were conducted. We evaluated the impact of pharmacological agents utilized in the treatment of CLL, including rituximab, bendamustine, fludarabine, cyclophosphamide, ibrutinib, chlorambucil, venetoclax, melphalan, and intravenous immunoglobulin, as well as the patients’ responses to these therapies (full or partial remission). The independent variables in this study were the patients’ demographic, clinical, and laboratory findings, while the dependent variable was prognosis (mortality).

### 2.3. CLL Diagnosis and Scoring Systems

Diagnosis of CLL was established based on B lymphocytosis with CD5 positivity (>5000/mm^3^ in peripheral blood) and a prolymphocyte ratio in peripheral blood of less than 55%, with clonality confirmed by flow cytometry. The Rai and Binet scoring systems were employed to assess the stages of CLL patients included in the study (Table 1).

### 2.4. Statistical Analysis

Descriptive statistics are presented as numbers (*n*), percentages (%), means (±standard deviations), and minimum–maximum values. Continuous independent variables were categorized into two groups by establishing cut-off values via ROC analysis. The cut-off values identified were 69.5 years for age, 12.7 g/dL for hemoglobin, 27.55 mm^3^ for leukocytes, 204.5 mm^3^ for platelets, 22.01 mm^3^ for lymphocytes, 4.43 mm^3^ for neutrophils, 13.5 mm/h for sedimentation rate, 14.5 U/L for ALT, 16.5 U/L for AST, 0.82 g/dL for creatinine, 45.5 g/dL for albumin, 195 U/L for LDH, and 0.48 mg/dL for uric acid. Survival analysis (Kaplan–Meier analysis) was conducted using variables identified as significant via a chi-square (Fisher’s Exact Test) assessment of prognostic factors. Statistical analyses were conducted using the Statistical Package for the Social Sciences version 22 (SPSS Inc., Chicago, IL, USA), with a significance threshold set at *p* < 0.05.

## 3. Results

A total of 178 patients were evaluated, and the mean age was 70.37 ± 11.49 years (range: 40 to 95), with 46.1% (*n* = 82) of the patients being female. According to the Binet staging system, 59.1% (*n* = 104) were classified as Stage A, 29.5% (*n* = 52) were classified as Stage B, and 11.4% (*n* = 20) were classified as Stage C. According to the Rai staging system, 51.7% of the patients (*n* = 91) corresponded to Stage 0, 8.5% (*n* = 15) corresponded to Stage 1, 23.9% (*n* = 42) corresponded to Stage 2, 11.9% (*n* = 21) corresponded to Stage 3, and 4.0% (*n* = 7) corresponded to Stage 4. Of the patients, 41.7% (*n* = 73) received treatment, with 3.4% (*n* = 6) of these being considered unresponsive to therapy. Additionally, 7.3% (*n* = 13) had another malignancy; 30.7% (*n* = 54) had splenomegaly; 37.5% (*n* = 66) had lymph node involvement; 18.8% (*n* = 31) had 17p deletion; 6.1% (*n* = 10) had Trisomy 13; and 26.7% (*n* = 44) had 13q deletions. The direct Coombs test was positive for 11.9% (*n* = 21) of the patients, and 4.5% (*n* = 8) of the patients died (Table 2). The factors affecting prognosis (mortality) were also evaluated, revealing that higher Binet Stages, being treated, the presence of splenomegaly, a positive Direct Coombs test, 17p deletion, and elevated levels of creatinine, leukocytes, and lymphocytes, as well as low platelet counts (Table 3), were associated with increased mortality and found to be statistically significant (Table 4). Among the 73 patients who received treatment, 56 (76.7%) were treated with rituximab; 41 patients (56.2%) received bendamustine; 18 patients (24.7%) were given ibrutinib; 15 patients (20.5%) received venetoclax; 12 patients (16.4%) were treated with chlorambucil; 11 patients (15.1%) received cyclophosphamide; 6 patients (8.2%) were administered IVIg; 5 patients (6.8%) received fludarabine; and 1 patient (1.4%) was treated with melphalan. Complete remission was achieved in 31 patients (42.5%), while partial remission was achieved in 36 patients (49.3%), and 6 patients (8.2%) were deemed unresponsive to treatment (Table 5). Seven of the eight patients who died were given rituximab, and one was given a mixture of rituximab and bendamustine. High Binet and Rai stages, the presence of 17p deletion, and low hemoglobin levels were revealed to be statistically significant predictors of therapeutic success (Table 6). The results that were deemed significant in the univariate analysis lost their significance in the multivariate analysis (Table 7). According to Kaplan–Meier survival analysis, the median survival times were as follows: 45 months for Stage A, 29 months for Stage B, and 26 months for Stage C. According to Binet staging system, the median survival time was 49 months for patients who received treatment, 32 months for those who were not treated, and 37 months for patients with splenomegaly. The median survival times were determined to be 39 months for patients with splenomegaly, 23 months for patients with a positive direct Coombs test, and 40 months for those with a negative test. The median survival time was 36 months for patients with 17p deletion and 39 months for those without. Additionally, the median survival times were 37 months for patients with creatinine levels ≤ 0.82 g/dL and 31 months for those with levels > 0.82 g/dL; 48 months for patients with leukocyte counts ≤ 27.55 mm^3^ and 25 months for those with counts > 27.55 mm^3^; 50 months for patients with lymphocyte counts ≤ 22.01 mm^3^ and 24 months for those with counts > 22.01 mm^3^; and 37 months for patients with platelet counts ≤ 204.5 mm^3^ and 43 months for those with counts > 204.5 mm^3^ (Figure 1).

## 4. Discussion

Chronic lymphocytic leukemia (CLL) is the most common hematological cancer, with numerous studies having been conducted to predict prognosis. Over the years, genetic anomalies have been identified as key prognostic factors. Traditional staging systems like Binet and Rai have been replaced by the CLL-IPI (International Prognostic Index), emphasizing genetic aberrations. However, few studies have explored the effects of additional factors on prognostic scores. In this study, we conducted a multiparameter analysis to evaluate routine testing, treatment modalities, and staging methods in CLL prognosis. Age and gender significantly influence CLL prognosis. A large-scale study (*n* = 36,007) found that 60% of patients are male, with a median age of 65 years or older [19]. Another cohort study reported a median age of 60 years, with three-quarters of patients being male [20]. Our study’s mean age of 70 years aligns with these findings [21,22]. However, unlike some studies reporting better survival among women [23] and shorter survival among patients aged 70 and above [24], we found no significant associations between age, gender, and mortality, possibly due to our small sample size. These discrepancies highlight the potential influence of population characteristics, healthcare access, and genetic predisposition in determining outcomes. The Binet staging system remains a crucial tool for evaluating CLL prognosis. It classifies patients based on lymphoid organ involvement and cytopenias. In our study, 58.1% of the patients corresponded to Stage A, differing from another study that reported a proportion of 77% [25]. Similarly, 50.8% of our patients corresponded to Rai Stage 0, contrasting with studies showing only 9% [26]. These differences may be attributed to variations in early detection, healthcare infrastructure, and screening practices. Early-stage diagnoses may result from factors such as comorbidities, frequent medical visits, and advancements in healthcare. Splenomegaly and lymphadenopathy significantly impact prognosis and treatment decisions [27,28]. In our study, 30% of the patients had splenomegaly, and 37% had lymphadenopathy, figures that are lower than those given in other reports [29,30,31]. Variability in prevalence may arise from differences in study design, patient demographics, and healthcare access. The clinical implications of these findings suggest that early identification and management of lymphoid involvement could alter disease trajectories. Genetic aberrations play a vital role in CLL pathophysiology and prognosis. The del(13q) chromosomal deletion, associated with a favorable outcome, was present in 24.6% of our patients, consistent with previous studies [15,32,33]. Trisomy 12, found in 6% of our cases, has been reported in 15–20% of CLL patients [34,35] and is linked to thrombocytopenia and secondary malignancies [36,37]. The del(17p) mutation, associated with rapid disease progression and poor prognosis, was detected in 17% of our patients, similar to previous reports [38,39,40,41]. This mutation is linked to treatment resistance and reduced survival [42,43]. Our findings reinforce the notion that del(17p) is a significant predictor of poor outcomes in CLL. These results emphasize the necessity of genetic screening in routine clinical practice to refine risk stratification and personalize treatment approaches. Mortality in CLL is influenced by factors such as comorbidities, treatment settings, and socioeconomic status [44,45]. The 5-year survival rate is reported to be 85%, decreasing to 65% for those aged 80 or older [46]. Our mortality rate of 4.5% aligns with studies reporting 3.7–5.2% [47]. Higher Binet stages correlated with increased mortality, consistent with prior research [48]. The Rai staging system did not reveal significant mortality differences in our study. Binet staging has been found to better predict prognosis, particularly with high lymphocyte counts and low platelet levels [49], although some studies found no significant difference between staging systems [50]. Our results confirm that a high Binet stage and del(17p) mutation, along with advanced Rai stage and low hemoglobin levels, influence treatment response [51]. Anemia, associated with poor quality of life, transfusion dependency, and chemotherapy-related toxicity, is a key prognostic factor [52]. These findings underscore the clinical importance of integrating staging systems with molecular markers to carry out more accurate risk assessments. Absolute lymphocytosis and elevated white blood cell counts are characteristic of CLL. We found that a low platelet count and high leukocyte and lymphocyte counts correlated with increased mortality. While anemia and thrombocytopenia are established poor prognostic indicators [14,16], distinguishing between disease-related cytopenias and autoimmune causes is critical. High leukocyte counts at diagnosis hold prognostic value, but their significance at later disease stages remains unclear [53]. Our study reveals an association between elevated creatinine levels and mortality, underscoring the importance of renal function monitoring. CLL-related renal dysfunction can arise from direct infiltration, autoimmune damage, lymphadenopathy-induced obstruction, or treatment effects. Studies have reported renal impairment in 16% of CLL patients [54], often seen in relapsed or treatment-resistant cases [55]. These results highlight the need for early renal function assessments and potential nephroprotective strategies in CLL management. Rituximab, an anti-CD20 monoclonal antibody, is widely used in CLL treatment, though long-term remission is not guaranteed [56,57,58]. In our study, three-quarters of the patients received rituximab, mostly in combination with bendamustine. Ibrutinib, preferred for patients with genetic deletions, was used in one-quarter of our cohort, consistent with its role in high-risk, relapsed, or treatment-resistant CLL [59,60]. The widespread use of these therapies highlights their central role in modern CLL treatment paradigms. However, the emergence of resistance and treatment-related toxicities necessitates ongoing research into alternative targeted therapies. Autoimmune complications, particularly autoimmune hemolytic anemia, are common in CLL. We investigated the prevalence of direct Coombs test (DCT) positivity and its prognostic implications. In our study, 12% of the patients were DCT-positive, which can be compared to 18–23% in other studies [61] and 30% in retrospective analyses [62]. DCT positivity correlates with advanced disease, poor treatment response, and higher mortality. Studies from China and other cohorts confirm there is shorter survival among DCT-positive patients [63]. These findings suggest that DCT may serve as a valuable prognostic marker in CLL. The inclusion of autoimmune markers in prognostic algorithms may enhance risk stratification and guide therapeutic decisions. The strong prognostic value of direct Coombs positivity and 17p deletion identified in our study provides important opportunities for updating and advancing current CLL staging systems. To optimize the use of these prognostic factors in clinical practice, an integrated staging system could be used, providing clinicians with a more comprehensive framework for determining treatment decisions and frequency of follow-up. The use of markers as an adjunct to staging systems in routine clinical practice may improve risk stratification and contribute to personalized treatment approaches.

In summary, our study reinforces the significance of staging systems, genetic aberrations, and clinical parameters in CLL prognosis. While our findings align with the existing literature in many respects, variations highlight the complexity of CLL pathophysiology and the need for individualized treatment approaches. Future research should focus on integrating novel biomarkers, refining staging criteria, and exploring personalized treatment strategies to optimize patient outcomes.

## 5. Conclusions

In this study, the factors affecting the prognosis of CLL included a high Binet stage, being treated, the presence of splenomegaly, a positive direct Coombs test, 17p deletion, a low platelet count, high creatinine levels, and increased leukocyte and lymphocyte values, all of which were associated with increased mortality. With advancements in molecular genetics and the development of new drugs targeting specific abnormalities, the Binet and Rai staging systems alone are insufficient for fully characterizing prognostic classification in CLL. This situation prompts the need for new scoring systems for more accurate risk assessments, treatment decisions, and follow-ups. Our results indicate that the Binet staging system remains a valuable marker for both mortality and treatment response. Nonetheless, the 17p deletion remains the most significant negative prognostic factor, indicating a poor treatment response and survival in CLL. While modern treatment approaches such as chemoimmunotherapy are highly effective for most CLL patients, the findings from both the literature and our study highlight the need for further randomized clinical trials, particularly for genetically high-risk patients. Based on our study, CLL patients with a positive direct Coombs test and impaired renal function should be monitored more closely due to their poor prognosis, and future research should aim to establish the benefits of early treatment initiation for these patients.

### Limitations

One of the most significant limitations of this study was the inability to exclude mortality due to non-hematological malignancies. Additionally, because the data were collected retrospectively, comorbidities and other medications used by the patients were not recorded. Other limitations include the fact that the study was conducted at a single center and involved a relatively small number of patients. However, the study’s strengths include its use of multiple prognostic indicators in evaluating prognosis and the limited number of similar studies available, which add value to our findings.

## Figures and Tables

**Figure 1 diagnostics-15-00728-f001:**
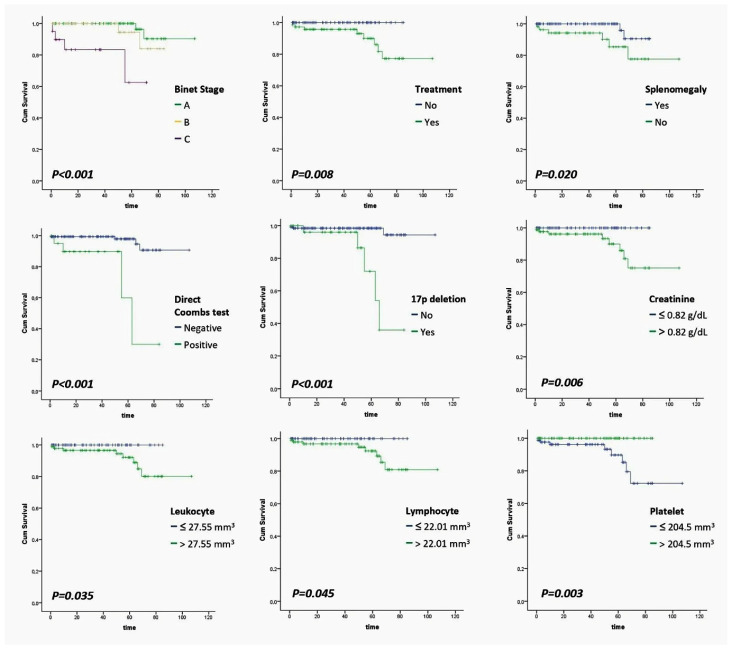
Kaplan–Meier survival graphics for different prognostic factors.

**Table 1 diagnostics-15-00728-t001:** CLL classification systems [14,16].

Rai			Binet		
Low Risk	Stage 0	Lymphocytosis	Low Risk	Stage A	<3 lymphoid areas
Intermediate Risk	Stage 1	Lymphadenopathy	Intermediate Risk	Stage B	≥3 lymphoid areas
	Stage 2	Spleen/liver enlarged			
High Risk	Stage 3	Hemoglobin < 11 g/dL	High Risk	Stage C	Hemoglobin < 10 g/dL
	Stage 4	Platelets < 100,000 mm^3^			Platelets < 100,000 mm^3^

**Table 2 diagnostics-15-00728-t002:** The demographic attributes of the study group.

(*n* = 178)	Total
Age, mean ± SD (min–max)	70.37 ± 11.49 (40–95)
Gender, *n* (%) (*n* = 178) Female Male	82 (46.1) 96 (53.9)
Binet Score, *n* (%) (*n* = 176) A B C	104 (59.1) 52 (29.5) 20 (11.4)
Rai Score, *n* (%) (*n* = 176) Low—0 Medium—1 Medium—2 High—3 High—4	91 (51.7) 15 (8.5) 42 (23.9) 21 (11.9) 7 (4.0)
Follow-up period, mean ± SD (min–max)	37.73 ± 26.85 (1–107)
Treatment, *n* (%) (*n* = 175) Yes No	73 (41.7) 102 (58.3)
Remission, *n* (%) (*n* = 73) Complete remission Partial remission No response	31 (42.5) 36 (49.3) 6 (8.2)
Malignity, *n* (%) (*n* = 177) Yes No	13 (7.3) 164 (92.7)
Splenomegaly, *n* (%) (*n* = 176) Yes No	54 (30.7) 122 (69.3)
Lymph node involvement, *n* (%) (*n* = 176) Yes No	66 (37.5) 110 (62.5)
D. Coombs, *n* (%) (*n* = 177) Positive Negative	21 (11.9) 156 (88.1)
17p deletion, *n* (%) (*n* = 165) Yes No	31 (18.8) 134 (81.2)
Trisomy 12, *n* (%) (*n* = 165) Yes No	10 (6.1) 155 (93.9)
13q deletion, *n* (%) (*n* = 165) Yes No	44 (26.7) 121 (73.3)
Prognosis (*n* = 178) Alive Deceased	170 (95.5) 8 (4.5)

SD: Standard deviation.

**Table 3 diagnostics-15-00728-t003:** Laboratory findings.

*n* = 178	Mean ± SD
Hemoglobin (g/dL)	12.70 ± 2.23
Leukocyte (mm^3^)	221.93 ± 88.35
Neutrophil (mm^3^)	8.18 ± 5.17
Lymphocyte (mm^3^)	44.32 ± 70.59
Reticulocyte	2.13 ± 7.37
Platelet (mm^3^)	213.10 ± 95.61
IGG	10.05 ± 4.76
Sedimentation (mm/h)	21.29 ± 20.68
ALT (U/L)	16.06 ± 9.56
AST (U/L)	18.37 ± 8.16
LDH (U/L)	221.93 ± 88.35
Creatinine (g/dL)	1.16 ± 4.07
Albumin (g/dL)	44.91 ± 3.74
Uric Acid (mg/dL)	9.16 ± 2.02
Total Bilirubin (mg/dL)	0.55 ± 0.36
Direct Bilirubin (mg/dL)	0.46 ± 3.14
Total Protein (mg/dL)	69.73 ± 5.92

Aspartate Aminotransferase (AST); Alanine Aminotransferase (ALT); Lactate Dehydrogenase (LDH); immunoglobulin G (IgG); SD: standard deviation.

**Table 4 diagnostics-15-00728-t004:** Comparison between independent variables (with cut-off values) and prognosis.

*n* (%)		Mortality		*p* *
		No	Yes	
Age	69.5≥	73 (98.6)	1 (1.4)	0.088
	>69.5	97 (93.3)	7 (6.7)	
Gender	Female	81 (98.8)	1 (1.2)	0.071
	Male	89 (92.7)	7 (7.3)	
Binet Scores	A	102 (98.1)	2 (1.9)	0.005
	B	50 (96.2)	2 (3.8)	
	C	16 (80)	4 (20)	
Rai Scores	Low	88 (97.8)	2 (2.2)	0.056
	Medium	56 (96.6)	2 (3.4)	
	High	24 (85.7)	4 (14.3)	
Treatment	Yes	65 (89)	8 (11)	0.001
	No	102 (100)	0 (0)	
Remission	Complete	30 (96.8)	1 (3.2)	0.192
	Partial	30 (83.3)	6 (16.7)	
	No response	5 (83.3)	1 (16.7)	
Malignity	Yes	11 (84.6)	2 (15.4)	0.109
	No	158 (96.3)	6 (3.7)	
Splenomegaly	Yes	48 (88.9)	6 (11.1)	0.011
	No	120 (98.4)	2 (1.6)	
Lymph node involvement	Yes	61 (92.4)	5 (7.6)	0.153
	No	107 (97.3)	3 (2.7)	
Direct Coombs test	Positive	17 (81.0)	4 (19.0)	0.008
	Negative	152 (97.4)	4 (2.6)	
17p deletion	Yes	26 (83.9)	5 (16.1)	0.008
	No	131 (97.8)	3 (2.2)	
Trisomy 12	Yes	10 (100.0)	0 (0.0)	0.461
	No	147 (94.8)	8 (5.2)	
13q deletion	Yes	42 (95.5)	2 (4.5)	0.913
	No	115 (95.0)	6 (5.0)	
Hemoglobin	12.7≥	77 (92.8)	6 (7.2)	0.103
	>12.7	92 (97.9)	2 (2.1)	
Leukocyte	27.55≥	83 (100.0)	0 (0.0)	0.007
	>27.55	86 (91.5)	8 (8.5)	
Platelet	204.5≥	78 (90.7)	8 (9.3)	0.003
	>204.5	91 (100.0)	0 (0.0)	
Lymphocyte	22.01≥	81 (100.0)	0 (0.0)	0.008
	>22.01	88 (91.7)	8 (8.3)	
Neutrophil	4.43≥	88 (96.7)	3 (3.3)	0.487
	>4.43	81 (94.2)	5 (5.8)	
Sedimentation	13.5≥	85 (95.5)	4 (4.5)	0.987
	>13.5	84 (95.5)	4 (4.5)	
ALT	14.5≥	85 (94.4)	5 (5.6)	0.721
	>14.5	84 (96.6)	3 (3.4)	
AST	16.5≥	80 (97.6)	2 (2.4)	0.289
	>16.5	89 (93.7)	6 (6.3)	
Creatinine	0.82≥	89 (100.0)	0 (0.0)	0.003
	>0.82	80 (90.9)	8 (9.1)	
Albumin	45.5≥	84 (94.4)	5 (5.6)	0.720
	>45.5	85 (96.6)	3 (3.4)	
LDH	195≥	84 (97.7)	2 (2.3)	0.279
	>195	85 (93.4)	6 (6.6)	
Uric acid	4.95≥	85 (97.7)	2 (2.3)	0.279
	>4.95	85 (93.4)	6 (6.6)	
Total Bilirubin	0.48≥	90 (94.7)	5 (5.3)	0.725
	>0.48	80 (96.4)	3 (3.6)	

Aspartate Aminotransferase (AST); Alanine Aminotransferase (ALT); Lactate Dehydrogenase (LDH); *: chi-square analysis (Fisher’s Exact Test).

**Table 5 diagnostics-15-00728-t005:** Evaluation of patients diagnosed with CLL who had been receiving drug therapy.

No	Age	Gender	Treatment	Remission	Prognosis
1	73	M	R, B	Complete	Alive
2	70	M	I	Complete	Alive
3	78	M	R, C, P, B, V	Partial	Alive
4	80	M	R, B	Complete	Alive
5	84	F	I	Partial	Deceased
6	80	F	I, IVIG	Partial	Alive
7	86	F	K	-	Alive
8	75	F	F	Complete	Alive
9	85	F	I	Complete	Alive
10	72	F	K, R, B, I	Partial	Alive
11	76	M	R, B	Complete	Alive
12	57	M	R, F, C	Complete	Alive
13	58	M	R, B	Complete	Alive
14	84	F	R, B	Partial	Alive
15	58	F	R, C, P	Complete	Alive
16	83	M	R, B	Complete	Alive
17	76	F	R, B	Partial	Alive
18	71	M	R, B	Complete	Alive
19	62	M	R-CHOP+R-ESHAP	Complete	Alive
20	34	M	R, F, C	Complete	Alive
21	70	F	R, C, P	Partial	Alive
22	78	F	R, B, I, V	Complete	Alive
23	72	M	R, B	Partial	Alive
24	81	F	R, B	Complete	Alive
25	63	M	R, B, I	Partial	Deceased
26	76	M	R, B	Partial	Alive
27	83	M	IVIG	Partial	Alive
28	51	M	R, B	Complete	Alive
29	80	M	R, B	Partial	Deceased
30	63	F	R, B, IVIG	Partial	Alive
31	81	M	R, B	Partial	Deceased
32	94	M	K	Partial	Alive
33	88	M	R, K, I	Partial	Deceased
34	79	M	R, B	Partial	Deceased
35	78	M	R, B	Complete	Alive
36	70	M	R, B	Complete	Deceased
37	73	M	IVIG	Partial	Alive
38	65	M	R, B	Complete	Alive
39	84	M	R, B, I, V	Partial	Alive
40	81	F	R, B, IVIG	Partial	Alive
41	85	M	R, K, M	Partial	Alive
42	95	M	R, B	Partial	Alive
43	92	M	I	Partial	Alive
44	61	M	R, B	Partial	Alive
45	71	F	IVIG	Partial	Alive
46	78	F	R, V	Partial	Alive
47	73	F	I	Complete	Alive
48	70	F	R, B	Partial	Alive
49	66	F	R, B	Partial	Alive
50	88	M	K	Partial	Alive
51	64	E	R, B	Partial	Alive
52	84	F	I	Complete	Alive
53	56	M	R-CHOP	Complete	Alive
54	75	M	R, B	Complete	Alive
55	75	F	K, I, V, IVIG	Complete	Alive
56	65	M	R, B	-	Alive
57	80	M	R, B, F, C	Partial	Alive
58	85	F	R, I	Complete	Alive
59	67	M	R, B	Partial	Alive
60	65	M	R, B	Partial	Alive
61	70	M	V	-	Alive
62	65	F	R, F, C, IVIG	Complete	Alive
63	50	F	R, B	Complete	Alive
64	75	F	I	Complete	Alive
65	75	M	R, V, CHOP	Complete	Alive
66	76	F	R, B	Complete	Alive
67	65	M	R, B	-	Alive
68	61	F	R, V, CHOP	Partial	Alive
69	87	M	I	Partial	Alive
70	83	F	R, K	Complete	Alive
71	80	M	R, K	Partial	Alive
72	67	F	R, K	-	Alive
73	71	M	R, B	-	Deceased

E: Male, F: female; R: rituximab, B: Bendamustine, F: fludarabine, C: cyclophosphamide, I: ibrutinib, K: chlorambucil, V: Venotoclasm, M: Melphalan, IVIG: intravenous immunoglobulin, CHOP: cyclophosphamide + adriamycin + vincristine + prednisolone, ESHAP: etoposide cisplatin cytarabine methylprednisolone.

**Table 6 diagnostics-15-00728-t006:** Evaluation of response to treatment for patients diagnosed with CLL (receiving treatment).

*n* (%)		Complete Remission	Partial Remission	No Response	*p* *
Age	69.5≥	9 (45.0)	8 (40.0)	3 (15.0)	0.356
	>69.5	22 (41.5)	28 (52.8)	3 (5.7)	
Gender	Female	14 (48.3)	13 (44.8)	2 (6.9)	0.711
	Male	17 (38.6)	23 (52.3)	4 (9.1)	
Binet Scores	A	11 (44.0)	14 (56.0)	0 (0.0)	0.026
	B	18 (54.5)	12 (36.4)	3 (9.1)	
	C	2 (13.3)	10 (66.7)	2 (20.0)	
Rai Scores	Low	7 (38.9)	11 (61.1)	0 (0.0)	0.011
	Medium	20 (60.6)	11 (33.3)	2 (6.1)	
	High	4 (18.2)	14 (63.3)	4 (18.2)	
Malignity	Yes	3 (33.3)	6 (66.7)	0 (0.0)	0.433
	No	28 (43.8)	30 (46.9)	6 (9.4)	
Splenomegaly	Yes	16 (43.2)	17 (45.9)	4 (10.8)	0.671
	No	15 (41.7)	19 (52.8)	2 (5.6)	
Lymph node involvement	Yes	22 (48.9)	19 (42.2)	4 (8.9)	0.301
	No	9 (32.1)	17 (60.7)	2 (7.1)	
Direct Coombs test	Positive	5 (31.2)	16 (66.7)	3 (12.5)	0.145
	Negative	26 (45.6)	17 (37.8)	3 (6.7)	
17p deletion	Yes	5 (20.8)	4 (57.1)	0 (0.0)	0.021
	No	25 (55.6)	29 (46.8)	6 (9.7)	
Trisomy 12	Yes	3 (42.9)	4 (57.1)	0 (0.0)	0.664
	No	27 (43.5)	29 (46.8)	6 (9.7)	
13q deletion	Yes	12 (44.4)	12 (44.4)	3 (11.1)	0.813
	No	18 (42.9)	21 (50.0)	3 (7.1)	
Hemoglobin	12.7≥	12 (24.0)	33 (66.0)	5 (10.0)	<0.001
	>12.7	19 (82.6)	3 (13.0)	1 (4.3)	
Leukocyte	27.55≥	7 (38.9)	10 (55.6)	1 (5.6)	0.792
	>27.55	24 (43.6)	26 (47.3)	5 (9.1)	
Platelet	204.5≥	19 (40.4)	22 (46.8)	6 (12.8)	0.164
	>204.5	12 (46.2)	14 (53.8)	0 (0.0)	
Lymphocyte	22.01≥	6 (37.5)	9 (56.2)	1 (6.2)	0.812
	>22.01	25 (43.9)	27 (47.4)	5 (8.8)	
Neutrophil	4.43≥	13 (34.2)	22 (57.9)	3 (7.9)	0.292
	>4.43	18 (51.4)	14 (40.0)	3 (8.6)	
Sedimentation	13.5≥	16 (44.4)	16 (44.4)	4 (11.1)	0.568
	>13.5	15 (40.5)	20 (54.1)	2 (5.4)	
ALT	14.5≥	14 (31.8)	26 (59.1)	4 (9.1)	0.074
	>14.5	17 (58.6)	10 (34.5)	2 (6.9)	
AST	16.5≥	10 (31.2)	18 (56.2)	4 (12.5)	0.173
	>16.5	21 (51.2)	18 (43.9)	2 (4.9)	
Creatinine	0.82≥	15 (50.0)	13 (43.3)	2 (6.7)	0.549
	>0.82	16 (37.2)	23 (53.5)	4 (9.3)	
Albumin	45.5≥	21 (43.8)	22 (45.8)	5 (10.4)	0.543
	>45.5	10 (40.0)	14 (56.0)	1 (4.0)	
LDH	195≥	14 (45.2)	13 (41.9)	4 (12.9)	0.346
	>195	17 (40.5)	23 (54.8)	2 (4.8)	
Uric acid	4.95≥	18 (56.2)	11 (34.4)	3 (9.4)	0.074
	>4.95	13 (31.7)	25 (61.0)	3 (7.3)	
Total Bilirubin	0.48≥	17 (42.5)	18 (45.0)	5 (12.5)	0.316
	>0.48	14 (42.4)	18 (54.5)	1 (3.0)	

Aspartate Aminotransferase (AST); Alanine Aminotransferase (ALT); Lactate Dehydrogenase (LDH); *: chi-square analysis (Fisher’s Exact Test).

**Table 7 diagnostics-15-00728-t007:** Cox regression analysis for mortality.

		*p*	Odds Ratio (95% Confidence Limits)
Binet Scores	A	0.252 0.319	1.000
	B		2.997 (0.359–3.394)
	C		3.501 (0.298–4.117)
Treatment	Yes		1.000
	No		
Splenomegaly	No	0.473	1.000
	Yes		2.170 (0.262–7.977)
Direct Coombs test	Negative	0.161	1.000
	Positive		6.672 (0.469–9.492)
17p deletion	No	0.280	1.000
	Yes		3.060 (0.403–3.347)
Leukocyte	27.55≥	0.984	1.000
	>27.55		1.108 (0.000–1.340)
Platelet	>204.5	0.798	1.000
	204.5≥		0.010 (0.000–5.199)
Lymphocyte	22.01≥	0.950	1.000
	>22.01		2.957 (0.000–6.591)
Creatinine	0.82≥	0.863	1.000
	>0.82		2.956 (0.000–5.208)

The first variables are the reference category.

## Data Availability

The data that support the findings of this study are available from the corresponding author upon reasonable request.

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
