# Peer review of "Chronic Lymphocytic Leukemia: Investigation of Survival and Prognostic Factors with Drug-Related Remission"

_diagnostics, 2025, doi:10.3390/diagnostics15060728_

Round 1
Reviewer 1 Report
Comments and Suggestions for Authors
I have reviewed the manuscript. Although it displays some interesting results, the manuscript needs to extensively modified.
1) In table 5, I would advise authors to use universal abbreviations for treatment modalities. Furthermore, I suggest them to group them (e.g., chemoimmunotherapy vs. venetoclax combinations vs. BTKis)
2) The authors should reanalyze their response and toxicity data comparing these 3 treatment groups (chemoimmunotherapy / venetoclax combinations / BTKis).
3) There are so many patients receiving chemoimmunotherapy as frontline treatment, as we all know that they are not generally recommended to be used (at least in high- and middle-income countries). This needs to be further discussed. Maybe the authors would like to divide the study period into two as 2015-2020 and 2020-2025 and compare the treatment distributions and etc. between these timeframes.
4) The causes of death should be shared. For example, we experienced COVID-19 pandemic during the study period, so, how many patients died due to COVID-19? These details should be shared and discussed.
5) The authors should perform uni- and multivariate analysis in order to detect independent factors on survival.
6) The discussion is repetitive and too long. It should be more concise.
Comments on the Quality of English LanguageThere are typos and grammatical errors throughout the manuscript that need to be checked and corrected.
Author Response
Comment 1
1) In table 5, I would advise authors to use universal abbreviations for treatment modalities. Furthermore, I suggest them to group them (e.g., chemoimmunotherapy vs. venetoclax combinations vs. BTKis)
Response 1:
Thank you for your suggestion. However, we believe that it would be more scientifically correct to prescribe each drug individually in our study of chronic lymphocytic leukemia rather than classifying them. The reason we prefer this approach is that it provides more useful data for future further analysis and studies.
Comment 2
2) The authors should reanalyze their response and toxicity data comparing these 3 treatment groups (chemoimmunotherapy / venetoclax combinations / BTKis).
Response 2:
Thank you for your request for a comparative analysis of the toxicity data of the drug combinations in our study. We would like to clarify important methodological issues in this context: The primary aim of our study was to assess the impact of multifactorial patient characteristics on prognosis. It was difficult to compare the efficacy or toxicity of drug treatments with each other, which was one of the aims of our study design. This is because comparing treatment modalities in terms of toxicity and efficacy requires a comprehensive study design that takes into account treatment doses, duration, duration of use, concomitant medications and patient characteristics, as well as prospective studies.
In addition, treatment decisions were made depending on the clinical characteristics, concomitant diseases and disease stage of the patients. Therefore, the drug treatment data in our study were presented only as descriptive statistics. The reason for this is that our study was not designed to compare the efficacy or toxicity of these treatments. We believe that within these methodological limitations, it would be more appropriate to present some results as descriptive statistics. Your suggestion will help us to develop a structured approach to the assessment of toxicity in the prospective studies we are planning.
Comment 3
3) There are so many patients receiving chemoimmunotherapy as frontline treatment, as we all know that they are not generally recommended to be used (at least in high- and middle-income countries). This needs to be further discussed. Maybe the authors would like to divide the study period into two as 2015-2020 and 2020-2025 and compare the treatment distributions and etc. between these timeframes.
Response 3:
Thank you for your valuable observations on the change in the use of chemoimmunotherapy in the treatment of chronic lymphocytic leukemia over time. We believe this is primarily due to the impact of important clinical factors such as patient risk profiles, comorbidities, deletion and mutation status, and patient age. However, treatment preferences at study sites are also influenced by factors such as patient preference, drug access policies and reimbursement constraints.
Despite the widespread use of new targeted therapies, chemoimmunotherapy is still an important component of CLL treatment. It remains a good option for selected patient groups. Chemoimmunotherapy, which remains a treatment option in current guidelines, continues to be used in low-income countries for appropriate patient profiles due to the difficulty of access and high cost of new treatments. Although new treatment options have increased over time in our study, one important point should be emphasized: The same patient may have received both chemoimmunotherapy and new generation targeted agents at different stages of treatment. The final prognosis of patients reflects the cumulative effect of all the treatments they have received. Considering only the first or last treatment may lead to misleading results. With our retrospective dataset, it does not seem methodologically possible to reliably isolate the prognostic impact of treatment type and to analyze these complex treatment histories reliably.
Comment 4
4) The causes of death should be shared. For example, we experienced COVID-19 pandemic during the study period, so, how many patients died due to COVID-19? These details should be shared and discussed.
Response 4:
As a retrospective study, the scope of the study's data collection was limited to the initially determined research objectives. Access to reliable and complete information about the causes of death is not possible during long follow-up periods for patients.
The collection and analysis of COVID-19-related mortality data is a sensitive category requiring special permission from health authorities. Detailed analysis of the causes of death, especially a limited focus on the COVID-19 period, goes beyond the main research question of our study and will lead us to an area that is not directly related to our original hypothesis.
Thank you again for your valuable suggestion. However, due to the above-mentioned limitations, it is unfortunately not possible to include a detailed analysis of the causes of death and COVID-19-related mortality data in our current study. This issue is a valuable research question for future studies to be planned with appropriate permissions and methodological design.
Comment 5
The authors should perform uni- and multivariate analysis in order to detect independent factors on survival.
Response 5:
Univariate analyses of survival were performed as part of the study (Table 4). With your valuable suggestions, a multivariate survival analysis (Cox regression analysis) was performed with the statistically significant values of the univariate analysis, and new results were added to the article (Table 7).
Comment 6
The discussion is repetitive and too long. It should be more concise.
Response 6:
When we checked our manuscript, we realized that you were absolutely right in your advice. In line with your request, we removed unnecessary repetitions in the discussion section and made it a little more fluent.
Reviewer 2 Report
Comments and Suggestions for Authors
This manuscript is dedicated to assessment of clinical parameters which can improve risk stratification strategies for CLL patients. Authors collected extensive clinical data for 178 patients over the course of ten years, and analyzed how a variety of parameters affects patient survival. Such clinical data is valuable on its own, although authors mostly confirm the existing practices for CLL risk assessment. I have no major concerns about provided analysis, but for now it lacks some deeper analysis and doesn’t improve existing understanding of CLL staging systems. It can be improved to make a full use of provided data, specifically providing analysis for treated patients separately would greatly improve significance of this study. There are several questions which can be further analyzed in this study:
1) How different are treatment outcomes for patients treated with rituximab, bendamustine, ibrutinib, venetoclax, chlorambucil, or cyclophosphamide? Are any of clinical parameters are associated with particular treatment outcome?
2) Clinical parameters were analyzed for all patients, including those that didn’t receive treatment. Patients who weren’t treated apparently didn’t have disease progression, so some of identified parameters may be a consequence of disease progression, rather a predictor for it. For example, hemoglobin levels, Coombs test and thrombocytopenia may result from advances disease stage and not be an independent factors. Authors should investigate how parameters associated with increased mortality are associated with survival specifically in treated patients.
3) Authors state that Direct Coombs positivity or 17p deletion are important prognostic factors. However, it is unclear how exactly those parameters might improve Binet or Rai staging systems? Are Direct Coombs positivity or 17p deletion more beneficial for stratification of lower or higher risk patients? For example, can Direct Coombs positivity improve predictions for intermediate or high risk patients? Some discussion on possible implementation of those parameters in staging system would also improve the manuscript.
Some minor issues:
1) Direct Coombs test is extensively discussed in the manuscript, but is not at all mentioned in the Introduction. Consider moving some sentences about it from Discussion to Introduction.
2) The firts part of Discussion (lines 172-250) largely restates the Introduction, does not discuss the actual study results and should be shortened and combined with the Introduction.
3) Tables need descriptions, especially descriptions for abbreviations, and units for measurements, such as Hemoglobin, Leukocyte, etc.
Author Response
Comment 1
1) How different are treatment outcomes for patients treated with rituximab, bendamustine, ibrutinib, venetoclax, chlorambucil, or cyclophosphamide? Are any of clinical parameters are associated with particular treatment outcome?
Response 1:
Thank you for your comments on the treatment outcomes of the various drugs used in the treatment of chronic lymphocytic leukemia (CLL). The retrospective nature of our study imposes important methodological limitations when comparing treatment efficacy. Treatment selection in CLL is often based on factors such as patient age, performance status, comorbidities, genetic/cytogenetic characteristics and disease stage. However, many patients receive more than one treatment during the course of their disease. Complex treatment sequences such as first-line treatment, second-line treatment, bridging therapies and palliative treatments make it difficult to distinguish which treatment influences the outcome. This can lead to serious errors in the interpretation of results regarding the efficacy or prognostic impact of the different treatment regimens. Therefore, descriptive statistics are provided to assess outcomes on a patient-specific basis. We examined the impact of other demographic, clinical, laboratory and genetic factors on treatment outcomes and prognosis and the results are presented in Tables 4 and 6.
Comment 2
2) Clinical parameters were analyzed for all patients, including those that didn’t receive treatment. Patients who weren’t treated apparently didn’t have disease progression, so some of identified parameters may be a consequence of disease progression, rather a predictor for it. For example, hemoglobin levels, Coombs test and thrombocytopenia may result from advances disease stage and not be an independent factors. Authors should investigate how parameters associated with increased mortality are associated with survival specifically in treated patients.
Response 2:
We thank you for your suggestion for a comprehensive evaluation of laboratory findings and other parameters that may influence prognosis in chronic lymphocytic leukemia. Based on similar suggestions in Reviewer 1 comment 5, we applied Cox regression analysis in our study for simultaneous evaluation of multiple variables that may influence prognosis. We present the results in Table 7. The results that we considered significant in individual analyses lost their significance in multiple analyses.
Comment 3
3) Authors state that Direct Coombs positivity or 17p deletion are important prognostic factors. However, it is unclear how exactly those parameters might improve Binet or Rai staging systems? Are Direct Coombs positivity or 17p deletion more beneficial for stratification of lower or higher risk patients? For example, can Direct Coombs positivity improve predictions for intermediate or high risk patients? Some discussion on possible implementation of those parameters in staging system would also improve the manuscript.
Response 3:
Thank you for your comment on the prognostic value of direct Coombs positivity and 17p deletion. Your valuable suggestion on how these parameters can be integrated into existing staging systems has enriched our study.
It is assumed that a comprehensive risk scoring system with clinical staging (Binet/Rai) + biological markers (17p del, Direct Coombs) + additional laboratory parameters can be developed with the help of our study and the results of other studies in this area.
In clinical practice, however, 17p deletion is considered a factor that determines the choice of treatment regardless of stage, and direct Coombs positivity is considered a factor that requires more frequent follow-up, especially in early-stage patients.
Some minor issues:
1) Direct Coombs test is extensively discussed in the manuscript, but is not at all mentioned in the Introduction. Consider moving some sentences about it from Discussion to Introduction.
Response: Necessary arrangements have been made in line with your request.
2) The firts part of Discussion (lines 172-250) largely restates the Introduction, does not discuss the actual study results and should be shortened and combined with the Introduction.
Response: Necessary arrangements have been made in line with your request.
3) Tables need descriptions, especially descriptions for abbreviations, and units for measurements, such as Hemoglobin, Leukocyte, etc.
Response: Necessary arrangements have been made in line with your request.
Round 2
Reviewer 1 Report
Comments and Suggestions for Authors
I have reviewed the revision. The manuscript is improved when compared to the original submission.
Reviewer 2 Report
Comments and Suggestions for Authors
Authors have adressed all major issues.